# A feasibility study evaluating the uptake, effectiveness and acceptability of routine screening of pregnant migrants for latent tuberculosis infection in antenatal care: a research protocol

A Rahman,[1] Shakila Thangaratinam,[2] Andrew Copas [iD],[3] D Zenner,[4] Peter J White,[5,6] Chris Griffiths [iD],[4] Ibrahim Abubakar [iD],[7] Christine McCourt,[8] Heinke Kunst[9]

For numbered affiliations see end of article.

**Correspondence to**
A Rahman;
ananna.rahman1@nhs.net

## ABSTRACT

**Introduction** Globally, tuberculosis (TB) is a leading cause of death in women of reproductive age and there is high risk of reactivation of latent tuberculosis infection (LTBI) in pregnancy. The uptake of routine screening of migrants for LTBI in the UK in primary care is low. Antenatal care is a novel setting which could improve uptake and can lend insight into the feasibility and acceptability of offering opt-out screening for LTBI.

**Methods and analysis** This is an observational feasibility study with a nested qualitative component. The setting will be the antenatal clinics in three hospitals in East London, UK . Inclusion criteria are pregnant migrant women aged 16–35 years attending antenatal clinics who are from countries with a TB incidence of greater than 150/100 000 including sub-Saharan Africa, and who have been in the UK for less than 5 years. Participants will be offered LTBI screening with an opt-out interferon gamma release assay blood test, and be invited to complete a questionnaire. Both participants and healthcare providers will be invited to participate in semistructured interviews or focus groups to evaluate understanding, feasibility and acceptability of routine opt-out LTBI screening. The primary analysis will focus on estimating the uptake of the screening programme along with the corresponding 95% CI. Secondary analysis will focus on estimating the test positivity. Qualitative analysis will evaluate the acceptability of offering routine opt-out LTBI screening to participants and healthcare providers.

**Ethics and dissemination** The study has received the following approvals: Health Research Authority (IRAS 247388) and National Health Service Ethics Committee (19/LO/0557). The results will be made available locally to antenatal clinics and primary care physicians, nationally to NHS England and Public Health England and internationally through conferences and journals.

**Trial registration number** NCT04098341.

## Strengths and limitations of this study

► The study uses a novel approach of tackling a complex problem of low uptake of latent tuberculosis infection (LTBI) screening in migrants by using an opt-out method and a novel setting: antenatal care.
► The study creates new education and training tools for healthcare professionals working in antenatal care.
► Our findings will provide a greater understanding of the acceptability of LTBI screening among pregnant migrant women and healthcare professionals.
► As this is an observational study, we are unable to demonstrate causality from our results.
► Not being able to interview women who decline LTBI screening may reduce the validity of our findings.

## INTRODUCTION
### Context
Tuberculosis (TB) remains a significant global health problem affecting an estimated 10 million people worldwide in 2019 leading to 1.4 million deaths.[1] TB is one of the leading causes of death in women of reproductive age (15–45 years).[2] In 2018, an estimated 3.2 million women globally were infected with TB and almost half a million women died from TB.[3] Indirect maternal deaths account for 28% of total maternal deaths, of which 15%–35% are due to TB.[2]

The WHO defines latent tuberculosis infection (LTBI) as a 'state of persistent immune response stimulation by *Mycobacterium Tuberculosis* antigens without evidence of clinically manifested active TB'.[1] A quarter of the world's population is estimated to have LTBI.[4] Individuals with LTBI have no signs

and symptoms of active TB but remain at risk of developing active TB in their lifetime. LTBI acts as a reservoir for active TB and TB elimination requires strategies for LTBI control.[5] The risk of reactivation of LTBI is higher in pregnancy.[6] This risk may be due to T-cell suppression and reduced interferon-gamma production.[7]

In low TB incidence countries, TB transmission is limited and most active cases of TB occur due to reactivation of LTBI imported from high incidence settings.[8] Uptake of LTBI screening in primary care is low.[9 10] Antenatal care is a new setting for LTBI screening and understanding the factors affecting the feasibility and acceptability of LTBI screening in this setting are first steps towards developing effective interventions to improving LTBI screening uptake.

## Current knowledge

In women of childbearing age (16–45 years), TB is one of three leading causes of death globally.[4] Diagnosis of TB in pregnancy is often delayed as pregnancy can mask some of the clinical manifestations of TB.[11 12] TB in pregnancy is associated with poor perinatal, foetal and maternal outcomes.[13 14]

The UK has one of the highest TB incidence rates in Western Europe. The incidence of TB among those born outside the UK is 14 times higher at 39.0 per 100 000 population and accounting for 74% of all new cases of TB in England in 2019.[15] Public Health England's TB migrant health guide strategy recommends migrant screening for LTBI in high incidence areas in England such as the London Boroughs of Tower Hamlets, Newham and Waltham Forrest.[16 17]

The London Borough of Newham was spearheading a large-scale LTBI screening programme in anticipation of the national programme. A total of 20 905 LTBI tests were reported between July 2014 and June 2017 across England with nearly half of the tests taking place in Newham.[9 10] Between April 2015 and June 2016, 5622 eligible migrants in England were offered an LTBI test, 2904 (51%) of whom attended for the test.[9]

Effective screening for LTBI is key to reducing TB incidence in the UK. There is good evidence that screening and treatment of LTBI is a cost-effective intervention that significantly reduces the risk of developing active disease and the risk of onward transmission.[18 19] The national LTBI migrant screening programme has been rolled out but there is insufficient evidence on the best setting for uptake of LTBI screening.

There is limited qualitative research about the acceptability to women of LTBI screening in pregnancy. Reasons for low uptake may be due to stigma of having active TB or fear of a positive test result affecting their immigration status. An opt-out approach to LTBI screening may normalise the process and has the potential to reduce barriers such as stigma, as well as practical barriers.[20]

Provider knowledge and understanding of the risks of TB, screening and treatment can be a major predictor of successful management of TB.[21] Data from a local LTBI screening programme have highlighted that offer of screening varies among general practitioner (GP) practices indicating that healthcare provider knowledge and attitude may influence offer of screening.[22]

Evaluating the impact of healthcare provider training to improve TB management has mainly been performed in low-income countries and there are only a few rigorous TB training evaluation studies available.[21] E-learning modules use pretraining and post-training tests to evaluate acquired knowledge. A GP E-learning module has been developed by the national TB charity 'TB Alert' to enhance knowledge of GPs responsible for screening and treatment of LTBI but the effectiveness of the module has not been formally evaluated.

## Rationale for LTBI screening in antenatal care

Pregnancy can predispose to reactivation of LTBI and diagnosis can be delayed due to reduced awareness among healthcare providers and reluctance to investigate non-specific TB symptoms by chest radiography.[23] Risks of LTBI reactivation and delays in diagnosis of TB can be mitigated by screening an at-risk pregnant migrant population for LTBI. A simple clinical algorithm recommended by the WHO based on the absence of current cough, fever, weight loss and night sweats can help to exclude active TB disease. Moreover, healthcare professionals will have a higher index of suspicion for active TB in interferon gamma release assay (IGRA) positive pregnant migrant women presenting with symptoms suggestive of TB, thus preventing a delay in diagnosis.[24]

Pregnant migrants may not be accessing routine healthcare and often do not have a GP. Antenatal care may therefore be a key opportunity to assess the woman's health and screen for TB. Antenatal care provides an opportunity for health promotion such as advocating GP registration and is a time when parents may be particularly receptive to public health information and promotion.

Routine opt-out testing has proven effective for other diseases (HIV/Hepatitis B, C).[25] Factors affecting successful uptake of screening programmes include how the test is offered, by whom, to whom and in what setting.[26] Pregnant women screened for HIV during pregnancy perceived routine opt-out HIV testing as beneficial for both women and their unborn babies.[26] Globally, some countries offer routine screening for TB in pregnancy mainly through symptom screen and sputum examination.[19]

LTBI screening for migrants from high TB incidence countries in antenatal care has shown high uptake in the USA, but feasibility of LTBI screening in antenatal clinics in the UK has not been evaluated.[27]

## Research hypothesis and aims

We hypothesise that offering routine opt-out LTBI screening to an at-risk pregnant migrant population in antenatal care will be feasible and acceptable to pregnant migrant women and healthcare providers.

To test our hypothesis, we will assess the uptake, feasibility and acceptability of screening an at-risk pregnant migrant population for LTBI at routine antenatal booking visits in secondary care, using opt-out IGRA testing. The results from this feasibility study will allow us to develop a definitive large-scale cluster randomised controlled trial (RCT) evaluating the effectiveness of an LTBI screening in antenatal care, the effectiveness of interventions used to maximise migrant screening for LTBI in pregnancy and to increase uptake of LTBI treatment postpartum.

## METHODS AND ANALYSIS
### Study protocol
This is a prospective observational feasibility study with nested qualitative research which will take place in antenatal booking clinics of three hospitals in East London (The Royal London Hospital, Newham University Hospital and Whipps Cross University Hospital). The study started on 29 April 2019 and the first participant was recruited on 3 July 2019. The study is due to finish on 31 May 2022.

Educational and training tools will be developed before the study begins. Healthcare providers involved in antenatal care will be asked to complete an E-learning module on active TB/LTBI, which has been developed by the study team, along with the national TB charity (TB Alert) and the Royal College of Midwives.

Study participants will enter the cohort when they attend the antenatal clinic for their booking appointment, after they meet inclusion criteria (table 1). Midwives will counsel and offer LTBI screening as an opt-out IGRA blood test alongside other routine investigations for blood-borne viruses at the initial booking appointment. The study will assume valid implied consent for participation if women undertake an IGRA test at the time it is offered by the midwife on an opt-out basis. Participants will be given a Participant Information Sheet by the midwife at this appointment detailing the study. Routine blood tests, including IGRA, will be taken by phlebotomists based in antenatal care.

At the time of offer of LTBI screening, we will record routine clinical data of all eligible pregnant migrant women including those who do not accept screening. Data on age, ethnicity, year of entry to the UK, pre-existing medical conditions and antenatal history which is routinely recorded in the medical notes will be collected.

All eligible pregnant migrant women will be screened for active TB by their midwives using a standardised symptom assessment questionnaire that includes the WHO recommended TB symptoms screen during their initial booking appointment. Study participants with a positive IGRA blood test will then undergo screening for active TB using the WHO recommended TB symptoms screen at 20 weeks, 30–34 weeks, delivery and postpartum. Data on symptoms of active TB will be collected at each time point (figure 1).

Participants will leave the study 6 weeks postdelivery or at the time of miscarriage if they have had a miscarriage.

Study participants with a positive IGRA blood test will be referred to the local TB clinic (if screened at The Royal London Hospital or Whipps Cross University Hospital) or to their GP (if screened in Newham University Hospital). TB clinics or GPs will review these individuals and initiate LTBI treatment according to local protocols.

All eligible pregnant women will be asked to complete a short questionnaire on acceptability of LTBI screening, knowledge about TB/LTBI and barriers to screening. At the end of pregnancy, women will be asked to complete the same questionnaire to compare the perception and knowledge of active TB/LTBI before and after the screening intervention. Trained research personnel will obtain written informed consent from the participant for the questionnaire.

We have used the Standard Protocol Items: Recommendations for Interventional Trials reporting guidelines for this paper.[28]

### Outcomes
Our primary outcomes are (i) the uptake of screening for LTBI in antenatal care assessed by the proportion of eligible migrant women offered a test who accepted LTBI screening, and (ii) the offer of IGRA blood test screening by healthcare providers assessed by the proportion of migrant women eligible for screening who were offered an IGRA test.

Secondary outcomes are as follows: rates of LTBI and active TB identified in pregnant migrant women during the study period, time to diagnosis, understanding and acceptability of LTBI screening and acceptability of interventions to increase screening uptake, perceived facilitators and barriers influencing uptake of LTBI screening and treatment uptake postpartum, increase in knowledge and awareness about active TB/LTBI among pregnant migrant women and healthcare providers and estimation of some of the parameters required for evaluation of cost-effectiveness of LTBI screening in antenatal care compared with primary care.

Process outcomes of the study are the numbers of eligible participants and screening acceptance rate, proportion of eligible pregnant migrant women who

| Table 1 | Inclusion and exclusion criteria |
| --- | --- |
| **Inclusion criteria** | **Exclusion criteria** |
| ► Pregnant migrant women aged 16–35 years AND<br>► from high TB incidence countries (incidence of TB of >150/100 000 including sub-Saharan Africa) AND<br>► who have been in the UK for less than 5 years | ► Previous history of TB or LTBI<br>► Individuals who are unable to consent<br>► Evidence of current active TB (based of history, examination, blood tests, chest X-ray findings or other radiological findings) |

LTBI, latent tuberculosis infection; TB, tuberculosis.

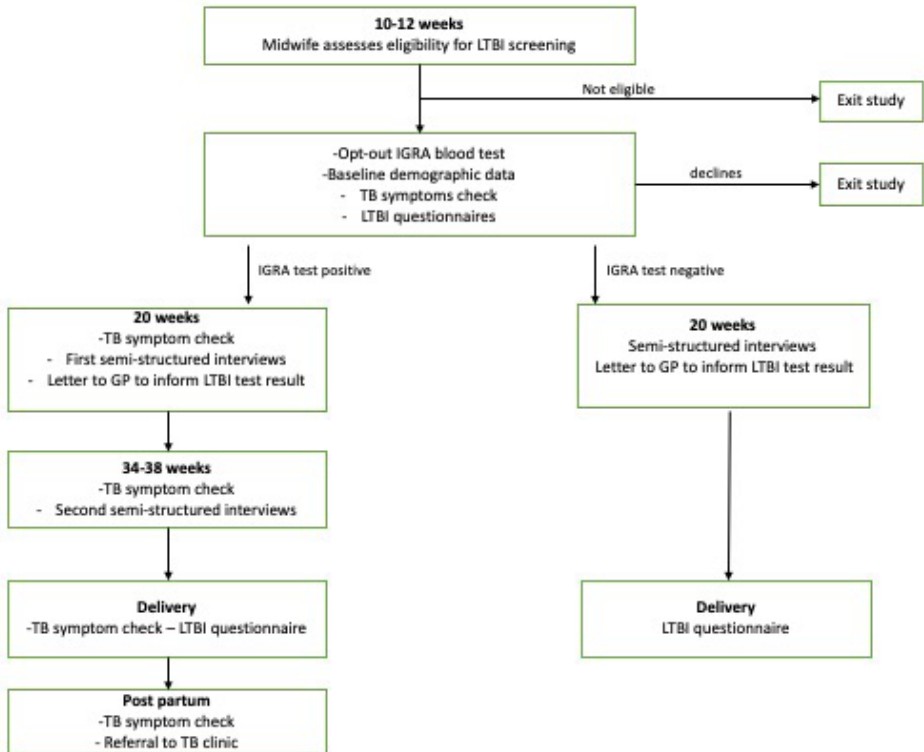

**Figure 1** Timeline of study project (assessment and follow-up of migrant women). IGRA, interferon gamma release assay; LTBI, latent tuberculosis infection; TB, tuberculosis.

were offered LTBI screening, views and experiences of participants on study recruitment methods, data collection methods and retention in the study and level of NHS support required for the proposed definitive cluster RCT.

## Patient and public Involvement

Healthwatch Newham conducted a survey to evaluate patient experiences of the LTBI screening programme in Newham, to identify the key factors that influence the uptake of screening, and to understand why patients decline screening. The results of this survey have influenced the design of this study, and migrants with LTBI have provided useful information about how LTBI screening could be better conducted. Evaluation of patient experiences demonstrated that migrants would like to be offered a LTBI test directly by their GP or nurse and that the test should be part of a general check-up. Our intervention has been designed to provide this by incorporating the offer of an IGRA test into routine antenatal care check-ups by midwives. The concept and the study design has been developed in close collaboration with TB Alert (UK TB charity), with the support of the East London Katherine Twining network PPI group (Katie's Team) and the Centre for Maternal and Child Health Research at City, University of London's service user panel and former TB/LTBI patients. PPI members felt that testing for LTBI as an opt-out approach is an acceptable intervention for pregnant migrant women.

## Sample size

A sample of 200 pregnant migrant women offered testing allows this study to estimate the screening uptake rate (key primary outcome for the feasibility study) with adequate precision across a range of possible values of the rate. If the uptake rate is 50% (at which precision is lowest), then this can be estimated within 6% either side, that is, a 95% CI of 44%–56%. If, however the rate is as high as 80% (or equivalently as low as 20%), then the rate can be estimated within 5%.

These precision calculations are based on the standard normal approximation and formula for a 95% CI for a proportion p based on a sample size n: $p \pm 1.96 \times sqrt[px(1-p)/n]$.

## Statistical analysis

The primary analysis will focus on estimating the uptake of the screening programme along with the corresponding 95% CI. Secondary analysis will focus on estimating test positivity. Associations between uptake and potential explanatory variables will be assessed using the $\chi^2$ test, and the strength of association will be presented as an OR with 95% CI. Identification of which characteristics are associated after adjusting for others will be performed using multiple logistic regression, and adjusted ORs will be presented.

## Nested qualitative research

Study participants will be invited to participate in semistructured interviews or focus groups to explore

acceptability of LTBI screening in antenatal care, understanding of LTBI among eligible pregnant women and healthcare providers, potential use of educational resources in each of these groups and potential barriers/facilitators to LTBI screening and treatment uptake.

A theoretical framework derived from the literature, survey and demographic data will be used to select a purposive sample to explore a range of relevant opinions and experiences. This will include interviewing women who have taken up screening as well as those who have not, or where this is not practicable, those within communities that might be offered screening. Sample size is guided by data saturation: for thematic analysis of semistructured interviews this is likely to occur between 10 and 40 participants and for focus groups 24–32 participants. Trained research personnel will obtain written informed consent from the participant for the semistructured interviews and focus groups.

Study participants will be invited to take part in two interviews. The first will take place early in the study (see figure 1) and will explore participants' understanding of LTBI, along with perceived acceptability of the study and intervention, participants' perceptions of their own risk of TB, their understanding of the prevention of TB and their views on the opt-out screening. The interview will also explore factors that influence participants' decision to be screened and suggestions for what might motivate them or other women to be screened, and their perspectives on the study data collection methods. Furthermore, participants' views and attitudes to LTBI treatment during or immediately after pregnancy will be assessed.

A second follow-up interview will take place towards the end of the study (see figure 1) with those participants who test IGRA positive to discuss their response to receiving a positive screening result, feelings around future treatment and explore what factors might encourage/discourage women from taking up treatment postpartum. Themes and concepts identified from the first set of interviews will inform the topics raised in the second interviews. This iterative approach will allow follow-up interviews to build on and explore further the participant experience, and to incorporate issues raised by other participants.

Women who decline participation in LTBI screening will be asked by recruiting midwives whether they consent for an independent researcher to contact them for an interview to explore their views. If few 'declining' women consent, up to three community-based focus groups will be conducted with migrant women of childbearing ages, and if appropriate men, in relevant populations to explore their awareness and their views about screening.

Semistructured interviews will also be conducted with 6–8 healthcare providers, including those who are involved in delivering the intervention, those who have expertise in managing pregnant women and local GPs to whom pregnant women may seek advice about screening and treatment for LTBI.

Two further focus groups with midwives, physicians and nurses, each involving around 8–12 participants will add a different perspective to that of the women. Their views and experiences on approaches to screening for TB/LTBI in antenatal care, along with perceived barriers/facilitators to LTBI screening and treatment, from a service or community perspective, will be explored.

Interview and focus group data will be analysed thematically, using constant comparison techniques, to identify, interpret and report patterns (themes) representing beliefs and experiences that participants share (or differ on) in relation to the research questions. The interviews and focus groups will also assess the views and experiences of participants and healthcare providers on study recruitment methods, data collection methods, facilitators and barriers to involvement, and compliance to study procedures.

## Data management

All study data will be managed according to the Clinical Effectiveness Group data management policy. Data will be entered directly onto a purpose-built database where possible (paper case report forms will be used as a backup if required).

Source data will be taken from the women's antenatal records and entered directly onto a database. Questionnaire data will be generated directly and then entered into the database.

The Investigator will ensure that patient anonymity is protected and maintained. They will also ensure that patient identities are protected from any unauthorised parties. Information with regard to study patients will be kept confidential and managed in accordance with the Data Protection Act, NHS Caldicott Guardian, The Research Governance Framework for Health and Social Care and Research Ethics Committee Approval.

The study will collect personal data and information about the participants either directly or from their clinical team. Routine clinical data will be entered onto a secure computer database, either by the research team or directly via a secure internet connection. The data will be pseudoanonymised. Any data processed by those outside the research team (research registrar, nurse or project coordinator) will be anonymised. All personal information obtained for the study will be held securely and treated as (strictly) confidential. All staff share the same duty of care to prevent unauthorised disclosure of personal information. No data that could be used to identify an individual will be published.

Transcripts from interviews and focus groups will be archived securely and audio-records destroyed securely following study closure in accordance with City, University of London's data management and retention policy. As all transcripts are deidentified at transcription

stage to ensure confidentiality, and personal data will be securely destroyed 1 year after study closure, no personal data will be included in archived records.

## Ethics and Dissemination

The study has received approval from The Health Research Authority (IRAS 247388) and London-City and East Research Ethics Committee (19/LO/0557). The study has been registered with clinicaltrials.gov (NCT04098341, preresults). The results will be made available locally to antenatal clinics and primary care physicians, nationally to NHS England and Public Health England and internationally through conferences and journals.

## DISCUSSION

Systematic national implementation of the LTBI screening programme is essential to achieving the aims of the collaborative strategy and support the WHO goal of TB elimination. The uptake of LTBI screening among migrants is low. This study seeks to provide patient-centred, migrant-inclusive evidence of the uptake, feasibility and acceptability of routine opt-out LTBI screening among pregnant migrants in antenatal care. It also seeks to understand potential facilitators and barriers from a healthcare provider perspective. We will assess whether this site of screening results in higher rates of LTBI screening uptake. The results of this study will inform the design of a cluster RCT trial evaluating the effectiveness of acceptable interventions to maximise migrant screening for LTBI in pregnancy, and to increase uptake of LTBI treatment postpartum.

## Author affiliations
[1] Blizard Institute, Queen Mary University of London, London, UK
[2] Institute of Translational Medicine, Birmingham University, Birmingham, UK
[3] Institute for Global Health, UCL, London, UK
[4] Institute of Population Health Sciences, Queen Mary University of London, London, UK
[5] MRC Centre for Global Infectious Disease Analysis and NIHR Health Protection Research Unit in Modelling and Health Economics, Imperial College, London, UK
[6] Modelling and Economics Unit, Public Health England, London, UK
[7] Tuberculosis Section, Centre for Infections, Health Protection Agency, London, UK
[8] Department of Midwifery and Child Health, City University London, London, UK
[9] Department of Respiratory Medicine, Queen Mary University of London, London, UK

Acknowledgements  We thank all the individuals who participated in this study. We also thank National Institute of Health Research (Funding reference: PB-PG-0317-20039), Health Research Authority (IRAS 247388) and London-City & East Research Ethics Committee (reference: 19/LO/0557).

Contributors  HK, ST, CMcC, AC, ZD, PJW, CG and IA designed the study and secured funding. AR trained healthcare providers and recruited participants for the study. AR wrote the first draft of the manuscript. All other authors reviewed drafts of this manuscript and commented upon them.

Funding  This study has been funded by National Institute of Health Research (NIHR), Research for patient benefit programme (Funding reference: PB-PG-0317–20039), Health Research Authority (IRAS 247388), City & East Research Ethics Committee (19/LO/0557).This study has been funded by National Institute of Health Research (NIHR), Research for patient benefit programme (Funding reference: PB-PG-0317–20039), Health Research Authority (IRAS 247388), City & East Research Ethics Committee (19/LO/0557). PJW acknowledges funding from the MRC Centre for Global Infectious Disease Analysis (grant number MR/

R015600/1); this award is jointly funded by the MRC and Foreign, Commonwealth and Development Office (FCDO) under the MRC/FCDO Concordat agreement and is also part of the European and Developing Countries Clinical Trials Partnership (EDCTP2) programme supported by the EU. PJW is also supported by the NIHR Health Protection Research Unit (HPRU) in Modelling and Health Economics, which is a partnership between Public Health England (PHE), Imperial College London, and LSHTM (grant code NIHR200908). The views expressed are those of the authors and not necessarily those of the UK Department of Health and Social Care, FCDO, EU, MRC, NIHR, or PHE.

**Patient and public involvement**  Patients and/or the public were involved in the design, or conduct, or reporting or dissemination plans of this research. Refer to the Methods section for further details.

**Patient consent for publication**  Not required.

**Provenance and peer review**  Not commissioned; externally peer reviewed.

**ORCID iDs**
Andrew Copas http://orcid.org/0000-0001-8968-5963
Chris Griffiths http://orcid.org/0000-0001-7935-8694
Ibrahim Abubakar http://orcid.org/0000-0002-0370-1430

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
