## [Reviewer comments · BMJ Open]

ARTICLE DETAILS

TITLE (PROVISIONAL)	A feasibility study evaluating the uptake, effectiveness and acceptability of routine screening of pregnant migrants for latent tuberculosis infection in antenatal care: a research protocol
AUTHORS	Rahman, A; Thangaratinam, Shakila; Copas, Andrew; Zenner, D; White, Peter; Griffiths, Chris; Abubakar, Ibrahim; McCourt, Christine; Kunst, Heinke

VERSION 1 – REVIEW

REVIEWER	Zhang, Hui China Center for Disease Control and Prevention, National Center for Tuberculosis Control and Prevention
REVIEW RETURNED	20-Dec-2021

GENERAL COMMENTS	It is suggested that the sample size calculation formula and the source of relevant parameters should be described in detail in the sample size calculation part.
---

REVIEWER	Rakesh, PuruShothama Amrita Institute of Medical Sciences and Research Centre
REVIEW RETURNED	04-Jan-2022

GENERAL COMMENTS	1. In strengths and limitations: There are sentences which are not strength/limitations- eg: The results will inform ways to increase uptake of LTBI screening in migrants in other settings such as primary care and Based on these results we will develop a definitive large-scale cluster randomized controlled trial to evaluate the effectiveness of LTBI screening in antenatal care.2. Context: The study is planned among migrant antenatal women, but the context is not set for it. It would be better if the researchers set the context focusing on Migrant antenatal women.3. It would be good if researchers describe the plan of management for a IGRA positive pregnant mother. That is important from an ethical point of view.4. Under the heading of research hypothesis - many things like - There is limited qualitative research about the acceptability to women of LTBI screening in pregnancy; Data from Newham's LTBI screening programme has highlighted that offer of screening varies amongst GP practices indicating that health care provider knowledge and attitude may influence offer of screening, entire paragraph on evaluating the impact of trainings, Factors affecting successful uptake of screening programmes include how the test
--

	is offered, by whom, to whom, and in what setting etc. etc. have been written. These sentences shall not fit under research hypothesis and aims. They may be appropriately described under current knowledge, rationale and justifications. 4. Kindly describe details about how the services will be offered and what all will be done- eg details of training of mid wives or health care providers, details of counselling & education, who will collect blood etc. 5. How the cost effectiveness will be estimated? kindly describe in the plan for analysis 6. In sample size, confidence interval is given as 54-66. It should have been 44-56. Kindly recheck 7. Kindly describe the study setting for a better understanding to the reader
--	---

VERSION 1 – AUTHOR RESPONSE

	First reviewer	
1.	It is suggested that the sample size calculation formula and the source of relevant parameters should be described in detail in the sample size calculation part.	We thank the reviewer for this comment. We have added the formula on page 13, sentence 1. This is a standard formula based on the variance of a sample proportion and the Normal approximation leading to the interval. We have considered a very wide range for the uptake rate, precisely because of our lack of knowledge prior to the study as to its likely value of this parameter. We have no specific justification for 6% precision, though it is adequate for our study. “These precision calculations are based on the standard Normal approximation and formula for a 95% confidence interval for a proportion p based on a sample size n: $p \pm 1.96 \times \sqrt{p \times (1-p) / n}$”
	Second reviewer	
1	In strengths and limitations: There are sentences which are not strength/limitations- eg: The results will inform ways to increase uptake of LTBI screening in migrants in other settings such as primary care and Based on these results we will develop a	We thank the reviewer for this comment. The ‘Strengths and limitations of this study’ section has been updated accordingly on page 3.

	definitive large-scale cluster randomized controlled trial to evaluate the effectiveness of LTBI screening in antenatal care.	
2	Context: The study is planned among migrant antenatal women, but the context is not set for it. It would be better if the researchers set the context focusing on Migrant antenatal women.	We thank the reviewer for this comment. The context section has been updated, page 4, sentence 2-4. “TB is one of the leading causes of death in women of reproductive age (15-45 years). In 2018, an estimated 3.2 million women globally were infected with TB and almost half a million women died from TB. Indirect maternal deaths account for 28% of total maternal deaths, of which 15-35% are due to TB”
3	It would be good if researchers describe the plan of management for a IGRA positive pregnant mother. That is important from an ethical point of view.	We thank the reviewer for this comment and have added a paragraph on page 10. “Study participants with a positive IGRA blood test will be referred to the local TB clinic (if screened at The Royal London Hospital or Whipps Cross University Hospital) or to their GP (if screened in Newham University Hospital). TB clinics or GPs will review these individuals and initiate LTBI treatment according to local protocols.”
4	Under the heading of research hypothesis - many things like - There is limited qualitative research about the acceptability to women of LTBI screening in pregnancy; Data from Newham’s LTBI screening programme has highlighted that offer of screening varies amongst GP practices indicating that health care provider knowledge and attitude may influence offer of screening, entire paragraph on evaluating the impact of trainings, Factors affecting successful uptake of screening programmes include how the test is offered, by whom, to whom, and in what setting etc. etc. have been written. These sentences shall not fit under research hypothesis and aims. They may be appropriately described under current knowledge, rationale and justifications.	We thank the reviewer for this comment and have updated the sections of current knowledge, rationale and justifications and research hypothesis on pages 5-8.

5	Kindly describe details about how the services will be offered and what all will be done- e.g. details of training of mid wives or health care providers, details of counselling & education, who will collect blood etc.	We thank the reviewer for this comment and have added a paragraph on page 9. “Study participants will enter the cohort when they attend the antenatal clinic for their booking appointment, after they meet inclusion criteria (Table 1). Midwives will counsel and offer LTBI screening as an opt-out IGRA blood test alongside other routine investigations for blood borne viruses at the initial booking appointment. The study will assume valid implied consent for participation if women undertake an IGRA test at the time it is offered by the midwife on an opt-out basis. Participants will be given a Participant Information Sheet by the midwife at this appointment detailing the study. Routine blood tests, including IGRA, will be taken by phlebotomists based in antenatal care”
6	How the cost effectiveness will be estimated? kindly describe in the plan for analysis	We thank the reviewer for this comment. The study will provide estimates of only some of the parameters required for health economic analysis and this health economic analysis is not part of the trial protocol. We have clarified this in the paper: “Secondary outcomes are: ... and estimation of some of the parameters required for evaluation of cost-effectiveness of LTBI screening in antenatal care compared to primary care”.
7	In sample size, confidence interval is given as 54-66. It should have been 44-56. Kindly recheck	We thank the reviewer for this comment and have updated the confidence interval to 44-56, on page 12, sentence 7. “If the uptake rate is 50% (at which precision is lowest) then this can be estimated within 6% either side, i.e., a 95% confidence interval of 44-56%.”
8	Kindly describe the study setting for a better understanding to the reader	We thank the reviewer for this comment and have added the study setting on page 8, sentence 5. “This is a prospective observational feasibility study with nested qualitative research which will take place in antenatal booking clinics of three hospitals in East London (The Royal London Hospital, Newham

		University Hospital and Whipps Cross University Hospital).”
--	--	---

VERSION 2 – REVIEW

REVIEWER	Rakesh, PuruShothama Amrita Institute of Medical Sciences and Research Centre
REVIEW RETURNED	02-Mar-2022

GENERAL COMMENTS	Thanks for addressing the questions.
--------------------------------------